# The Essential Oil of *Petroselinum crispum* (Mill) Fuss Seeds from Peru: Phytotoxic Activity and In Silico Evaluation on the Target Enzyme of the Glyphosate Herbicide

**DOI:** 10.3390/plants12122288

**Published:** 2023-06-12

**Authors:** Oscar Herrera-Calderon, Abdulrahman M. Saleh, Ammar A. Razzak Mahmood, Mohamed A. Khalaf, James Calva, Eddie Loyola-Gonzales, Freddy Emilio Tataje-Napuri, Haydee Chávez, José Santiago Almeida-Galindo, Javier Hernán Chavez-Espinoza, Josefa Bertha Pari-Olarte

**Affiliations:** 1Department of Pharmacology, Bromatology and Toxicology, Faculty of Pharmacy and Biochemistry, Universidad Nacional Mayor de San Marcos, Lima 15001, Peru; 2Pharmaceutical Medicinal Chemistry & Drug Design Department, Faculty of Pharmacy (Boys), Al-Azhar University, Cairo 11884, Egypt; abdo.saleh240@azhar.edu.eg; 3Department of Pharmaceutical Chemistry, College of Pharmacy, University of Baghdad, Bab-Almouadam, Baghdad 10001, Iraq; kubbaammar1963@gmail.com; 4Chemistry Department, College of Science, United Arab Emirates University, Al-Ain P.O. Box 15551, United Arab Emirates; 202170149@uaeu.ac.ae; 5Departamento de Química, Universidad Técnica Particular de Loja, Loja 1101608, Ecuador; jwcalva@utpl.edu.ec; 6Department of Pharmaceutical Science, Faculty of Pharmacy and Biochemistry, Universidad Nacional San Luis Gonzaga, Ica 11001, Peru; eddie.loyola@unica.edu.pe; 7Departamento de Ciencias Comunitarias de la Facultad de Odontología, Universidad Nacional San Luis Gonzaga, Ica 11001, Peru; freddy.tataje@unica.edu.pe; 8Department of Pharmaceutical Chemistry, Faculty of Pharmacy and Biochemistry, Universidad Nacional San Luis Gonzaga, Ica 11001, Peru; hchavez@unica.edu.pe (H.C.); javierchavez@unica.edu.pe (J.H.C.-E.); bertha.pari@unica.edu.pe (J.B.P.-O.); 9Department of Basic Sciences, Faculty of Human Medicine, Universidad Nacional San Luis Gonzaga, Ica 11001, Peru; santiago.almeida@unica.edu.pe

**Keywords:** aromatic plants, parsley, bioherbicide, volatile oil, essential oil, GC-MS, in silico

## Abstract

*Petroselinum crispum* (Mill) Fuss is an aromatic plant belonging to the *Apiaceae* family and used in gastronomy as a spice. Several studies have been developed in leaves but studies are limited in seeds, especially the essential oils obtained from seeds. The aim of this study was to determine the phytochemical profile of the volatile compounds of this essential oil by gas-chromatography–mass spectrometry (GC-MS) in order to evaluate its phytotoxic activity on *Lactuca sativa* seeds and to carry out an in silico analysis on the target enzyme of the herbicide glyphosate 5-enolpyruvylshikimate 3-phosphate synthase (EPSP). The essential oil was obtained by steam distillation for two hours and then was injected into a GC-MS, the phytotoxic assay was carried out on Lactuca seeds and the in silico evaluation on the EPSP synthase focused on the volatile compounds similar to glyphosate, docking analysis, and molecular dynamics to establish the protein–ligand stability of the most active molecule. The chromatographic analysis revealed 47 compounds, predominated by three compounds with the most abundant percentage in the total content (1,3,8-ρ-menthatriene (22.59%); apiole (22.41%); and β-phellandrene (15.02%)). The phytotoxic activity demonstrated that the essential oil had a high activity at 5% against *L. sativa* seed germination, inhibition of root length, and hypocotyl length, which is comparable to 2% glyphosate. The molecular docking on EPSP synthase revealed that trans-p-menth-6-en-2,8-diol had a high affinity with the enzyme EPSP synthase and a better stability during the molecular dynamic. According to the results, the essential oil of *P. crispum* seeds presented a phytotoxic activity and might be useful as a bioherbicide agent against weeds.

## 1. Introduction

Around the world, herbicide-resistant weeds represent a serious problem in food security with a current presence of 500 cases; the USA and Australia are the leading countries with a high number of herbicide-resistance cases, having reported 160 and 90 cases, respectively, followed by Canada, China, and Brazil [1]. In particular, the weeds that grow on crops such as maize, wheat, soybean, rice, canola, spring barley, and cotton have shown a high herbicide resistance [2]. Weeds are a type of plant that can rapidly reproduce and limit crop production by competing for several elements such as water, light, soil nutrients, and space for deep root systems, etc., [3]. An increase in weeds during agricultural production can trigger yield losses in crops and can depend on the type of weeds, crops, climate, weed density, weed emergence time, and others [4]. Additionally, these weeds might release a type of chemical substance called an allelopathic substance, which could negatively affect crops [5]. Despite the reduction management of weeds being a focus during the employment of synthetic herbicide, serious damage to the environment is produced due to its negative effects which include the acceleration of soil erosion, herbicide-resistant weeds, and contamination in vegetables and/or foods [2]. 

Bioherbicides are considered a powerful tool to combat weeds without causing any damage to the environment because they originate from nature sources and are less toxic than synthetic herbicides such as glyphosate [6]. Regarding glyphosate ((N-(phosphonomethyl) glycine)), this synthetic herbicide is sold at a low cost, but lately there have been concerns about its carcinogenic effect in humans [7]. Glyphosate biochemically works as a competitive inhibitor of phosphoenolpyruvate in the shikimic acid pathway, leading to the inhibition of the enzyme enolpyruvylshikimate-3-phosphate synthase (EPSP), a key enzyme in the generation of aromatic amino acids for plant growth [8]. Within the active site of EPSP synthase, several key residues play essential roles in catalysis and substrate binding. One of these critical residues is a conserved aspartate residue (Asp) and lysine (Lys), which aids in substrate binding and stabilizes the transition state during the reaction [9]. These active site residues work cooperatively to ensure efficient catalysis and precise substrate specificity, making EPSP synthase an attractive target for the development of herbicides and antimicrobial agents [10].

Certain bioherbicides have as their main chemical agent essential oils (EOs), which are also considered allelopathic substances [11]. EOs are volatile chemicals obtained by a distillation process or non-conventional methods (supercritical fluid CO_2_ extraction, microwaves, ultrasound, etc.) [12] and their composition is mainly based on monoterpenes and sesquiterpenes compounds, although other compounds including aliphatic and aromatic esters have also been reported in EOs. Currently, the use of EOs in sustainable agriculture is being widely investigated to produce potential bioherbicides, insecticides, antibacterial, antifungal, antiviral, nematicides, and other effects to avoid crop losses due to weeds [13]. In addition, an issue linked to organic farming is the use of chemical herbicides which open up the possibility of searching for phytocompounds such as EOs with a bioherbicidal potential. 

*Petroselinum crispum* (Mill) Fuss is a perennial herb (Figure 1), native to the Mediterranean region and used in the food, cosmetic, and pharmaceutical industries for its aromatic properties; it is commonly known as parsley and belongs to the *Apiacea* family [14]. It has been introduced and cultivated in many parts of the world due to its medicinal effects such as: antiseptic, stimulant, emmenagogue, antispasmodic, carminative, and stomachic effects, and its action on the uterine fiber [15,16,17]. On the other hand, the EO of *P. crispum* presents a high content of apiol [18], myrcene, 1,3,8-p-menthatriene, myristicin, β-phellandrene, and other terpenoids [19]. However, these chemical profiles can vary depending on several factors, mainly the origin, the extraction method, and the part of the plant used to obtain the EO [20]. The aim of this investigation was to determine the volatile compounds of *P. crispum* EO by gas-chromatography–mass spectrometry (GC-MS), in order to evaluate their phytotoxic activity on *Lactuca sativa* seeds and carry out an in silico analysis on EPSP synthase—the target enzyme of the herbicide glyphosate.

## 2. Results and Discussion

### 2.1. Chemical Characterization of EO of P. crispum Seeds

The obtained EO of *P. crispum* presented a light-yellow color with an extraction yield of 0.106%, a relative density of 0.9100 ± 0.02 g/mL at 20 °C, and a refractive index of 1.4910 at 20 °C. The chemical profile identified by GC-MS is reported in Table 1. The EO revealed 47 compounds (Table 1), 4 of which had unknown structures. The GC-MS identified the presence of 1,3,8-ρ-menthatriene as the major component with 22.59%, followed by apiole (22.41%) and β-phellandrene (15.02%). The studies reported about the EO of *P. crispum* seeds vary according to the precedence, type of extraction, pre-treatments, and environment factors. Piras et al. [21] compared the EOs from seeds obtained by supercritical extraction and hydrodistillation, during which apiol was the most abundant compound with 65.4% and 82.1%, respectively, followed by myristicin with 20.1% and 11.4%. Furthermore, the yield percentage was better in supercritical extraction than in hydrodistillation [21]. The EO from Morocco presented apiole (23.45%), α-pinene (18.98%), and β-pinene (15.6%) as the main volatile compounds [22]. The EO from Italy had as its main volatile component myristicin with 19.4%, followed by 1,3,8-p-menthatriene (17.7%), β-phellandrene (17.2%), and apiole (15.5%). In addition, this EO was composed of monoterpene hydrocarbons (52.9%) and its content of phenylpropanoids was 34.9%, similar to our findings [23]. In a study carried out in China regarding the EO obtained by microwave and hydrodistillation, the content of myristicin was 79.58% and 85.59%, respectively [24]. Other reports of an EO from Turkey revealed that 3-methoxy-γ-asarone was the major compound with 34.19%, followed by myristicin with 23.83%. The variability in composition observed in comparison with the other *P. crispum* EOs from seeds discussed above could be due to the collection season, edaphic and climate factors, the previous pretreatment during the obtention of the EO in this study, and whether the seeds were milled and placed immediately into the distillation apparatus. However, the difference in percentages between 1,3,8-ρ-menthatriene and apiole in the EO from Peru is not significant; this investigation is the first report of this type, contradicting those findings in which myristicin is the main component.

### 2.2. Phytotoxic Activity of the EO of P. crispum on Lactuca sativa Seeds

In Table 2, it can be noted that there was no significant difference in some of the parameters evaluated to determine the phytotoxic effect between *P. crispum* EO at 0.5% and 1% with glyphosate, in seed germination (*p* = 0.1587 and *p* = 0.9690, respectively) or root length (*p* = 0.0784 and *p* = 0.1796, respectively), but in hypocotyl length, there was a significant difference at 0.1%, 0.5%, and 1% (*p* = 0.0004, *p* = 0.0067, *p* = 0.0424, respectively). Figure 2A shows the normal germination of *L. sativa* seed treated with 0.1% DMSO and does not evidence any alteration in growth. In Figure 2B–D, treated with the *P. crispum* EO, a reduction in root length was observed depending on the concentration of the used EO, as well as the hypocotyl length reduction. Figure 2E shows the effect of glyphosate in the germination process with a reduction in root and hypocotyl length. 

The use of essential oils as promissory bioherbicides is generating much interest due to the toxicity in the environment produced by synthetic herbicides, which can cause serious damage to the ecosystem. In this study, glyphosate was used as a positive control, which is used as a synthetic herbicide that blocks the enzyme EPSP synthase and catalyzes the sixth step in the shikimic acid pathway, reducing aromatic amino acids such as phenylalanine, tyrosine, and tryptophan [25] and its phytotoxic activity is also presented with the inhibition of seed germination, root length, hypocotyl length, and the rate of root length/stem length depending on the concentration, while the negative control (0.1% DMSO) did not show any phytotoxic activity. Germination was inhibited at the highest concentration, being similar to the positive control (1% glyphosate). Currently, there are no reports about the phytotoxicity of *P. crispum* EO from seeds. However, during a study in which the plant organ used to obtain the EO was not revealed, *P. crispum* at 0.64 µg/mL reduced the germination at 37.1%, reduced the root length at 29.1%, and resulted in a total fresh weight reduction at 79.7% [26]. 

On the other hand, certain volatile terpenes have demonstrated phytotoxic activity against *L. sativa* seeds including: 1,8-cineole, sabinene, and α-pinene from *Vitex agnus-castus* [27]; p-cymene, γ-terpinene, thymol, carvacrol, and borneol from *Thymus eigii* [28]; pulegone [29], β-pinene, δ^3^-carene, and limonene from *Heterothalamus psiadioides* [30]; α-pinene, γ-terpinene, and p-cymene from *Eucalyptus grandis* [31]; α-pinene, and β-pinene from *Pinus brutia* [32]; eucalyptol, linalool and β-myrcene from *Artemisia absinthium* [33]; and 1,8-cineole, β-phellandrene, α-pinene from *Majorana hortensis* [34]. Several of these volatile compounds were reported in the GC-MS analysis (Table 1) but in low percentages, except for β-phellandrene which might synergize with the phytotoxic effect.

In addition, the phytotoxic activity of the EO might be linked to its content of oxygenated terpenoids. This conclusion is supported by several studies of EO with phytotoxic activity in which an oxygenated terpenoid represents the major component, such as eucalyptol in species of *Eucalyptus* [35], carvacrol in *Thymus proximus* [36], and menthol in *Mentha × piperita* [37]. However, in this study, oxygenated terpenoids are only represented by oxygenated monoterpenes (2.07%) and oxygenated sesquiterpenes (0.792%) at low percentages of the total content. Additionally, apiol at 80 mM has shown a reduction in growth of *Lemna paucicostata* seeds at 75% [38]. Subsequently, monoterpenes and sesquiterpenes extracted from EOs have shown phytotoxic effects, causing anatomical and physiological changes in plants; proposed mechanisms involve the accumulation of lipid globules in the cytoplasm, oxidative stress, a reduction in mitochondria, and an inhibition of DNA [39]. 

### 2.3. Oxidative Stress Markers in the Phytotoxic Activity of the EO of P. crispum on Lactuca sativa Seeds

In Figure 3, the markers of oxidative stress such as malondialdehyde (MDA) and superoxide dismutase (SOD) are observably increased in those groups treated with the EO and the positive control glyphosate. As shown in Figure 3, there was no significant difference between *P. crispum* at 1% (1.72 ± 0.07 µM g^−1^ DW) and glyphosate (1.72 ± 0.07 µM g^−1^ DW) in MDA. For SOD, *P. crispum* at 1% (0.98 ± 0.12 U mg^−1^ DW) and glyphosate (1.05 ± 0.1 U mg^−1^ DW) provide similar values. According to Đorđević et al. [40], lipid peroxidation represents a parameter of oxidative damage to molecules and cellular structures, especially those that contains lipids; in this research, MDA was evaluated and those seeds exposed to the EO at different concentrations showed an increased MDA at 1% overall. In addition, SOD is considered to be one of the main enzymatic systems to inhibit the stress generated by free radicals in plants [41]. Figure 3 shows that *L. sativa* seeds exposed to the EO increased their levels of SOD in response to the damage generated by the EO, similar to glyphosate. Several terpenoids such as α-pinene and β-pinene have shown an increase in MDA, catalase, SOD, peroxidase, and ascorbate as an action mechanism in phytotoxicity [42]. Certain species of plants have demonstrated phytotoxic activity via increased lipid peroxidation (MDA) such as the EOs of *Pogostemon benghalensis, Monarda didyma,* and *Artemisia scoparia* [43]. 

### 2.4. Molecular Similarity

Table 3 represents the molecular similarity data for a set of compounds. The table includes various molecular descriptors such as the A-LogP (predicted octanol–water partition coefficient), molecular weight, number of hydrogen bond acceptors (HBA), number of hydrogen bond donors (HBD), number of rotatable bonds, number of rings, number of aromatic rings, and minimum distance. Based on the given data, the top 11 compounds are marked as similar, while the remaining compounds are marked as dissimilar. The top 11 compounds have molecular weights ranging from 138.20 to 222.36, and A-LogP values ranging from 1.313 to 3.202. These values suggest that the compounds are relatively small, with a moderate lipophilicity. They also have a similar number of hydrogen bond acceptors and donors, with values ranging from 1 to 4 which indicate the ability to form good hydrogen bonds with the target site. The number of rotatable bonds is relatively low, ranging from 1 to 3, indicating a semi-flexible structure. These compounds also have a relatively low number of rings, ranging from 1 to 3, with only a few having aromatic rings. In contrast, the dissimilar compounds have molecular weights ranging from 132.202 to 238.28 and A-LogP values ranging from 2.772 to 4.939. These values suggest that these compounds are larger and more hydrophobic than similar compounds. Furthermore, most of these compounds have zero or very low values for HBA and HBD, indicating a lack of polar functional groups. These compounds also have a higher number of rotatable bonds and rings, ranging from 0 to 7, indicating a more flexible and complex structure. The minimum distance values shown in the table represent the similarity between each compound and a reference compound, the co-crystalized ligand 2AAY. The top 11 compounds have minimum distance values ranging from 2.07 to 3.11, indicating a high degree of similarity with the reference compound. In contrast, the dissimilar compounds have minimum distance values ranging from 3.5249 to 3.8803, indicating a low degree of similarity. In summary, the data in the table suggest that the top 11 compounds are similar in terms of their molecular properties, while the remaining compounds are dissimilar. The top 11 compounds have a moderate molecular weight and lipophilicity, with a similar number of polar functional groups, and a relatively rigid and simple structure. In contrast, the dissimilar compounds are larger and more hydrophobic, with fewer polar functional groups, and a more flexible and complex structure (Figure 4).

### 2.5. Docking Studies 

The most similar compounds were docked against the EPSP synthase target site to study their binding mode and the degree of affinity towards this target site. The co-crystalized ligand showed an energy binding of −6.55 kcal/mol to the EPSP synthase target site, indicating its binding mode. the co-crystalized ligand formed six hydrogen bonds with Arg27, Ser23, Lys22, Asp313, and Lys340, with distances of 1.86, 1.78, 5.82, 2.46, 1.72, and 1.84 Å (Figure 5).

In addition, cis-ρ-mentha-2,8-dien-1-ol had an energy binding of −5.47 kcal/mol to the EPSP synthase target site, and it formed two pi–alkyl interactions with Tyr200 and Lys340. It also had interactions with Asp313 and Lys22 through two hydrogen bonds with distances of 1.77 and 2.12 Å, as shown in Figure 6. On the other hand, terpineol had an energy binding of −5.69 kcal/mol to the EPSP synthase target site. It interacted with Lys22 and Asp313 through two hydrogen bonds with distances of 1.80 and 1.67 Å, respectively. Moreover, it formed a pi–alkyl interaction with Tyr200 (Figure 7).

The binding orientation of the cis-ρ-mentha-1(7),8-dien-2-ol had an affinity score of −5.66 kcal/mol against the EPSP synthase target site, and it formed three pi–alkyl interactions with Tyr200, Pro312, and Lys340. It also had interactions with Asp313 and Lys22 through two hydrogen bonds with distances of 1.83 and 1.93 Å, respectively, as shown in Figure 8. Meanwhile, the trans-p-mentha-8-thiol-3-one had an energy binding of −5.56 kcal/mol against the EPSP synthase target site. It formed two pi–alkyl interactions with Lys340 and Tyr200 and two hydrogen bonds with Lys22 and Lys340 with distances of 1.74 and 2.35 Å, respectively, as shown in Figure 9. 

Trans-p-menth-6-en-2,8-diol’s binding mechanism demonstrated an energy binding of −5.54 kcal/mol against the target site of EPSP synthase. Asp313, Lys340, and Lys22 were also interacted with by trans-p-menth-6-en-2,8-diol through three hydrogen bonds that were 1.75, 4.86, and 2.02 Å apart. (Figure 10). Additionally, apiole’s binding mode demonstrated an energy binding of −6.85 kcal/mol against the target site of EPSP synthase; it interacted with Asp313 and Tyr200 via three pi–alkyl, pi–pi, and pi–anion interactions; and it also interacted with Lys340 and Arg124 via two hydrogen bonds with distances of 2.43 and 2.12 Å (Figure 11).

Table 4 shows the DS, RMSD, and interactions in kcal/mol of each tested metabolite against the EPSP synthase target site, from which it can be concluded that apiole was the best molecule with a high affinity to the protein target −6.85 kcal/mol, similar to glyphosate at −6.55 kcal/mol. However, the compound trans-p-menth-6-en-2,8-diol had a greater number of hydrogen bonds than other compounds. Therefore, the molecular dynamic was carried out with this compound. The other values of the volatile compounds are presented in Appendix A.

According to the molecular docking results, trans-p-menth-6-en-2,8-diol, an oxygenated monoterpene of *P. crispum* EO, presented a high affinity for the EPSP synthase. In one study, several ligands were evaluated against EPSP synthase, and those with a high affinity for the enzyme had hydrogen-bond interactions with residues Lys23, Arg404, Arg105, Asp50, Gly101, Arg131, and Thr102. Furthermore, some of those residues were important in the interactions between phosphoenolpyruvate and glyphosate in the phosphoenolpyruvate-binding site [44]. These reports confirm that trans-p-menth-6-en-2,8-diol from *P. crispum* EO might be used as an herbicide in the future via its interactions with similar residues to those reported by Oliveira et al. [44] such as Asp313, Lys340, and Lys22.

### 2.6. Molecular Dynamic Simulation 

The study utilized molecular dynamics simulations to assess the stability and conformational changes of a protein–ligand complex. The root mean square deviation (RMSD) was used to evaluate the dynamic movements and conformational variations in both the protein and ligand, revealing a consistently low RMSD by the simulation. This indicates a high level of stability within both the apo and ligand-bound states. In order to identify the regions of the protein that exhibit flexibility during the simulation, the RMSF was calculated for each residue. Interestingly, no significant differences were noticed in the flexibility of any residues upon binding of the ligand. In order to assess the compactness of the complex, the radius of gyration (Rg) was calculated. The Rg values remained relatively constant throughout the simulation, indicating that the system remained compact. However, a slight increase in Rg was observed for the complex compared to the starting period. In order to investigate the interaction between the protein–ligand complex and the solvent, the solvent accessible surface area (SASA) was calculated. It was found that the protein had a lower SASA value at the end of the simulation, indicating a reduction in surface area and suggesting a more stable structure. Lastly, the study investigated the hydrogen bonding between the protein and ligand. It was observed that the protein and ligand could create up to four hydrogen bonds, which likely helped to keep the complex stable (Figure 12).

### 2.7. The Molecular Mechanics Poisson–Boltzmann Surface Area (MMPBSA)

The study used the MM/PBSA method to calculate the binding free energy between the protein–ligand complex for the last 20 ns of the MD production run at intervals of 100 ps. The MmPbSaStat.py script was employed to obtain the average binding free energy and its standard deviation/error from the output files obtained from g_mmpbsa. The results showed that the binding free energy between the ligand and protein was −28 KJ/mol, indicating a favorable interaction. The study also identified the contribution of individual protein residues towards binding free energy. The study discovered that the LYS22, PRO312, TRP337, and ARG386 residues of the protein had a larger contribution of more than −2 KJ/mol toward the interaction with the ligand. This was done by breaking down the overall binding free energy into the per residue contribution energy. These residues were identified as hotspot residues in the binding of the ligand to the protein. A figure illustrating these results is provided (Figure 13). In another investigation, certain residues of the enzyme had a positive contribution with the ligand, enhancing its interaction, which were Arg124, Asp313, Glu341, His385, Lys22, Gln171, Arg344, Arg386, and Gly196 [45]. 

## 3. Materials and Methods

### 3.1. Plant Material

A quantity of 10 kg of *P. crispum* aerial parts was collected from Ica, Peru, located at 458 masl (14°04′40.5″ S and 75°43′15.5″ W)) in October 2022 (Figure 1). The material plant was authenticated in the Herbarium of the Universidad Nacional Mayor de San Marcos (070-2022-USM-MHN). Seeds (1 kg) were selected, cleaned, and milled. Then, seeds were placed into a Clevenger apparatus to obtain the EO by steam distillation for 2 h. Finally, the EO was separated from the hydrolate by decantation, and anhydrous Na_2_S

O_4_ was added to purify the EO. Finally, the EO was kept in a sealed amber vial at 4 °C.

### 3.2. Identification of the Volatile Compounds by Gas Chromatography–Mass Spectrometry (GC–MS)

The volatile components of EO of *P. crispum* seeds were identified using an Agilent 6890N gas chromatograph coupled to an Agilent 5973 Mass Selective Detector operating in electron-ionization mode at 70 eV with a 5% diphenyl and 95% dimethylpolysiloxane capillary column (DB-5 MS, 30 m *×* 0.25 mm *×* 0.25 μm). The EO was diluted in CH_2_Cl_2_ of HPLC grade (1: 100 *v/v*). The chromatographic conditions were set according to a previous analysis protocol [46]. The identification of volatile components was based on a comparison of relative retention indices (RIs), mass spectra data (NIST–5 library), and published literature. Each retention index was calculated in comparison to a homologous series of n-alkanes C9–C25 (C9, 99% BHD purity and C10–C25, 99% Fluka purity). 

### 3.3. Phytotoxicity Assay on Lactuca sativa Seeds

Germination and growth bioassays were performed in Petri dishes (90 mm in diameter), with Whatman filter paper number 42. Commercial seeds of *Lactuca sativa* (Anasac ^®^, Batch N° 393368-61, distributed by Hortus S.A., Ate, Lima, Peru) were treated with different solutions of the EO solubilized in 0.1% DMSO at 0.1; 0.5; and 1.0%, respectively. The Petri dishes were then incubated at 20 °C for 5 days under a photoperiod of 12 h light. Several parameters were evaluated, including the number of germinated seeds, the root length, and the hypocotyl length. A solution of 0.1% DMSO was used as a negative control, and a 1% glyphosate solution (Fuego^®^, NeoAgro, Lima, Peru) was used as a positive control. Each Petri dish had twenty seeds, and five replicates were utilized [47].

### 3.4. Determination of Oxidative Stress Parameters in L. sativa 

The values of malondialdehyde (MDA) were measured using the reagent thiobarbituric acid. A total of 50 μL of *L. sativa* shoots homogenized in phosphate-buffered saline reacted with 200 μL 0.25% thiobarbituric acid solution (solution containing 10% trichloroacetic acid); then, these samples were heated at 95 °C for 30 min using a water bath. Next, samples were cooled to be centrifuged at 10,000× *g* for 10 min. The MDA content was measured using absorbance at 532 nm, and 155 mM^−1^ cm^−1^ was taken as the extinction coefficient. The formula used is:$$MDA=\frac{A}{155\;mM^{-1}cm^{-1}}$$
where A is the absorbance at 532 nm. 

Superoxide dismutase (SOD) was determined using a commercial kit (Sigma Chemical Co, Saint Louis, MA, USA) according to the manufacturer’s instructions. The SOD reaction was read on a microplate reader at 450 nm. Each assay was repeated in triplicate [48].

The formula used were:$$Percent\;inhibition\;\left(\%\right)=\frac{\left(\Delta \;OD_{s}\;\;min^{-1}-\Delta OD_{0}\;min^{-1}\right)}{\left(\Delta \;OD_{s}\;\;min^{-1}\right)}\times 100$$

Increase in absorbance (uninhibited) per min (ΔODs min^−1^) at 450 nm; inhibition of absorbance per min (ΔODo min^−1^) by the sample at 450 nm.
$$SOD\;activity\;\;in\;seeds=\frac{Percent\;inhibition\times DF}{\;50\%\times sample\;volume\;\left(mL\right)\times \;U\;mg^{-1}\times protein}$$

### 3.5. Method of Molecular Similarity

The molecular similarity calculation is a computational method that evaluates the similarity between two molecules based on their ligands’ structural and physicochemical properties, such as LogP, molecular weight, and distances between descriptors. In this study, six ligands with anticancer activity against an angiotensin-receptor blocker were analyzed for molecular similarity with glyphosate as a reference using Discovery studio software. The molecular properties analyzed included the rotatable bonds, cyclic rings, aromatic rings, hydrogen bond donors (HBD), hydrogen bond acceptors (HBA), partition coefficient (A-LogP), molecular weight (M.wt), and molecular fractional polar surface area (MFPSA) (Figure 4). 

### 3.6. Method of Docking Study

The AUTODOCK VINA v.1.2.0 software was used to test ten natural metabolites against the EPSPS target site. To generate the binding sites, the co-crystallized ligand within crystal protein (PDB code: 2AAY) obtained from the RCSB was used [49]. The targeted proteins were prepared by removing water molecules, performing preparation options, adding missing amino acids, correcting unfilled valence atoms, and minimizing the protein peptides energy by applying CHARMM force fields. The essential amino acids of the protein were selected and prepared for screening. The natural metabolites were prepared by drawing their 2D structures using Chem-Bio Draw Ultra17.0 and saving them in SDF file format. The saved files were then opened in AUTODOCK VINA v.1.2.0 software, and the ligands were protonated, and energy was minimized using an MMFF94 force field with 0.1 RMSD kcal/mol. The minimized structures were stored for molecular docking. Molecular docking was performed using docking algorithms, where the targeted pocket was held rigid, and the ligands were allowed to be flexible. Each molecule was allowed to produce twenty different interaction poses with the protein during the refinement. The docking scores (affinity interaction energy) of the best-fitted poses with the active site at the EPSP synthase target site were recorded, and the 3D orientation was generated using the Biovia Discovery Studio 2019 Visualizer [50].

### 3.7. Method of Molecular Dynamic Simulation and MMPBSA

Molecular dynamics (MD) simulation is a computational method that investigates the behavior of molecules and atoms by numerically solving the equations of motion of a system. The widely used Gromacs software was employed to perform the MD simulations. The protein–ligand complex was prepared using Chimera software, and topology and parameter files were generated using Gromacs. The system was solvated with water molecules and equilibrated with NVT ensembles (constant number of particles, volume, and temperature) and NPT ensembles (constant number of particles, pressure, and temperature), both of which were carried out at 300 K and 1 atm for 100 picoseconds (ps). The thermostat and barostat for the simulation that was run, respectively, were chosen to be the V-rescale and Parrinello–Rahman. Finally, the production run was carried out for 100 nanoseconds (ns) at 300 K with a 2-fs time step. The molecular mechanics Poisson–Boltzmann surface area (MMPBSA) method was utilized to calculate the binding free energy between the protein and ligand. The trajectory files were analyzed using Gromacs tools to determine the RMSD, RMSF, SASA, hydrogen bonds, total binding energy, and radius of gyration. These were calculated using Gromacs in 100 ns to understand the dynamic behavior of the protein–ligand complex and the effect of ligand binding on protein structure and stability. The obtained results provided insights into the stability and interactions of the protein–ligand complex, which can aid in drug discovery and design [51].

## 4. Conclusions

The EO from the seeds of *P. crispum* collected in Ica, Peru, was analyzed for the first time by GC-MS and found to contain a high content of terpene hydrocarbons and phenylpropenes. The most abundant volatile compound was 1,3,8-ρ-menthatriene (22.59%), followed by apiol (22.41%). This finding indicates the existence of chemotypes of *P. crispum* which may be useful to identify and assess the quality control of the EO of Peruvian *P. crispum* seeds. Moreover, the EO produced a phytotoxic effect on *L. sativa* seeds at 1%, similar to glyphosate. The inhibition of germination, root length, and hypocotyl length were achieved at 0.5% and 1% concentrations, and its involved mechanism could be inducing oxidative stress due to a high concentration of MDA and SOD found in *L. sativa* shoots. Regarding the in silico analysis, 11 volatile compounds were similar to glyphosate during the molecular similarity analysis. In addition, trans-p-menth-6-en-2,8-diol had a high affinity to the enzyme EPSP synthase with −5.54 kcal/mol; furthermore, it presented a stability with the enzyme residues, Lys22, Pro312, Trp337, and Arg386 being important residues in the energy contribution during the molecular dynamic. This investigation will lead to continued study of the EO as a potential bioherbicide to combat weeds, and the authors aim to carry out toxicological studies in vivo, as well as to determine the genotoxic effect to assess its safety profile.

## Figures and Tables

**Figure 1 plants-12-02288-f001:**
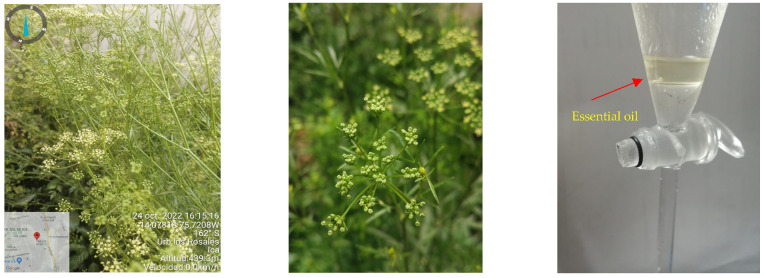
The EO of *P. crispum* seeds obtained by steam distillation.

**Figure 2 plants-12-02288-f002:**
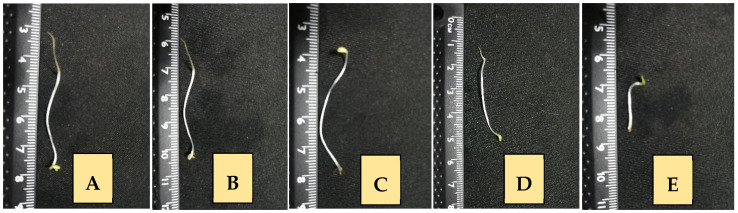
Phytotoxic activity of the EO of *P. crispum* on *Lactuca sativa* seeds. (**A**) Negative control (DMSO 0.1%); (**B**) *P. crispum* 0.1%; (**C**) *P. crispum* 0.5%; (**D**) *P. crispum* 1%; (**E**) glyphosate 1%.

**Figure 3 plants-12-02288-f003:**
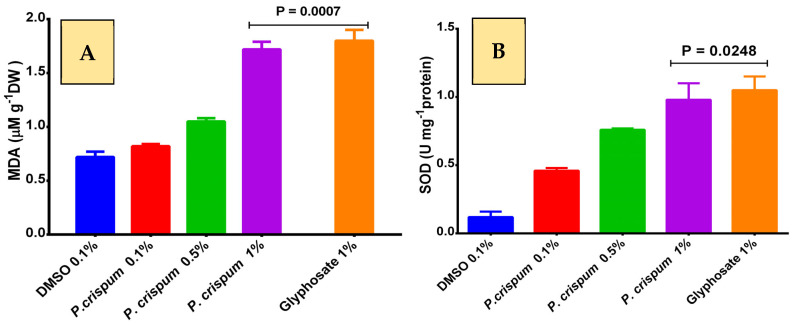
Oxidative stress evaluation in *L. sativa* shoots exposed to *P. crispum* (0.1–1%) and 1% glyphosate. (**A**) Malondialdehyde (MDA); (**B**) antioxidative enzyme superoxide dismutase (SOD). Results are expressed as the average of three determinations. *p* ≤ 0.05 is considered as significant (Tukey’s test).

**Figure 4 plants-12-02288-f004:**
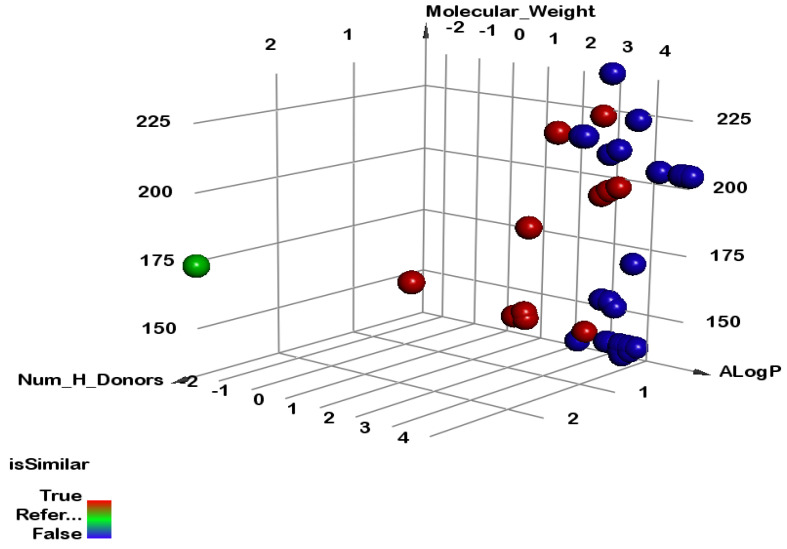
The similarity analysis between the volatile compounds of *P. crispum* and glyphosate. Green ball = reference ligand (glyphosate), red balls = similar ligands, blue balls = not similar ligands.

**Figure 5 plants-12-02288-f005:**
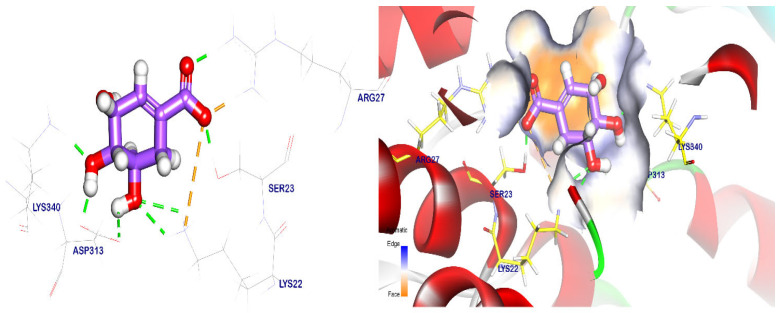
The co-crystalized ligand was placed back into the EPSP synthase target site and the green lines represent the hydrogen bonds while the purple lines represent pi interactions. The surface mapping revealed that the co-crystalized ligand occupied the active pocket of the EPSP synthase target site.

**Figure 6 plants-12-02288-f006:**
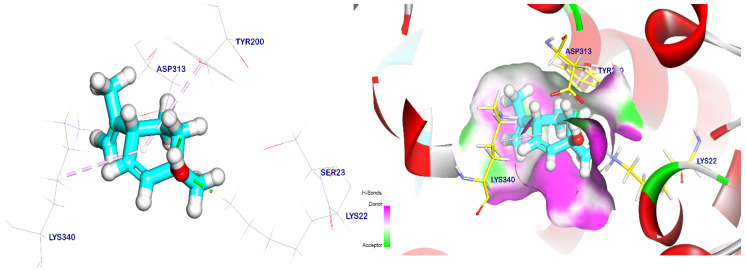
The EPSP synthase target site is occupied by cis-ρ-mentha-2,8-dien-1-ol. Hydrogen bonds are represented by green lines, while pi interactions are shown as purple lines. The surface mapping indicates that cis-ρ-mentha-2,8-dien-1-ol occupies the active pocket of the EPSP synthase target site.

**Figure 7 plants-12-02288-f007:**
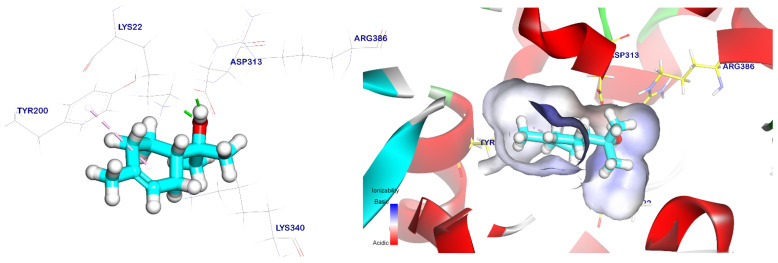
Terpineol docked in the EPSP synthase target site. Hydrogen bonds are represented by green lines, while pi interactions are shown as purple lines. The surface mapping indicates that terpineol occupies the active pocket of the EPSP synthase target site.

**Figure 8 plants-12-02288-f008:**
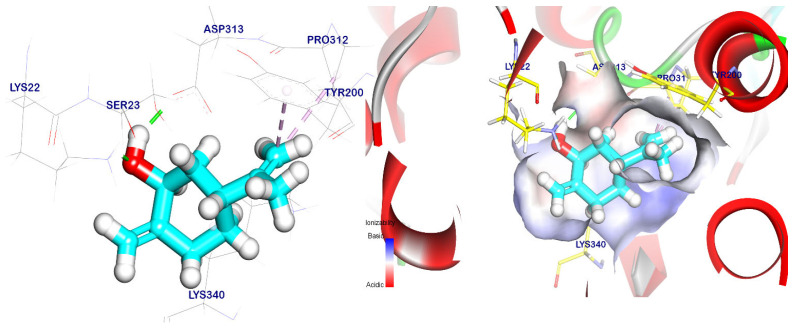
Cis-mentha-1(7),8-dien-2-ol docked in the EPSP synthase target site. Hydrogen bonds are represented by green lines, while pi interactions are shown as purple lines. Surface mapping reveals cis-mentha-1(7),8-dien-2-ol occupying the active pocket of the EPSP synthase target site.

**Figure 9 plants-12-02288-f009:**
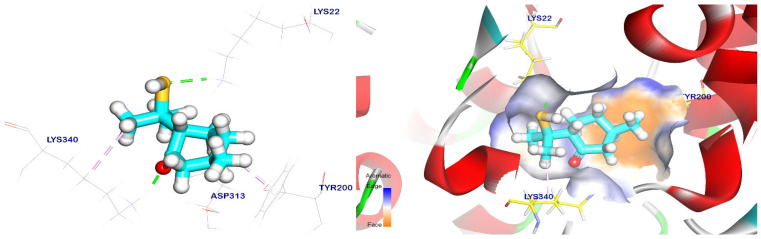
Trans-p-mentha-8-thiol-3-one docked in the EPSP synthase target site. Hydrogen bonds were represented by green lines while pi interactions were shown as purple lines. Surface mapping shows trans-p-mentha-8-thiol-3-one occupying the active pocket of the EPSP synthase target site.

**Figure 10 plants-12-02288-f010:**
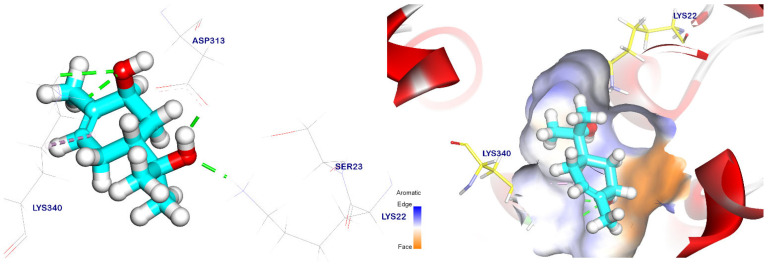
Surface mapping demonstrates that trans-p-menth-6-en-2,8-diol is docked in the active pocket of the EPSP synthase target site. Hydrogen bonds are represented by green lines, while pi interactions are shown as purple lines.

**Figure 11 plants-12-02288-f011:**
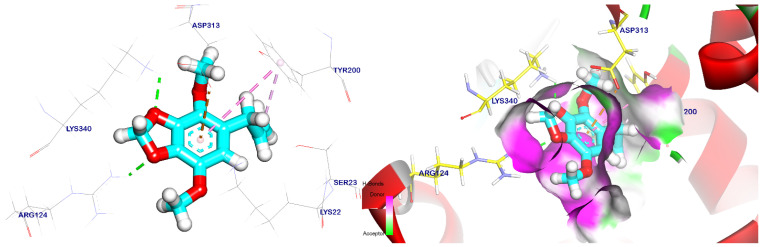
Apiole docked in the EPSP synthase target site. Hydrogen bonds are represented by green lines, while pi interactions are shown as purple lines, with surface mapping demonstrating apiole occupying the active pocket of the EPSP synthase target site.

**Figure 12 plants-12-02288-f012:**
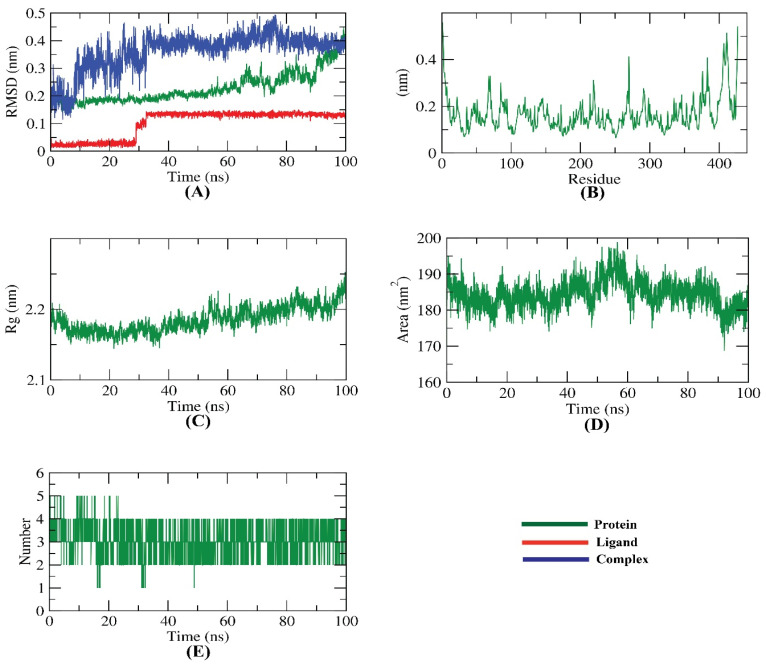
MD simulations of the EPSP synthase target site–trans-p-menth-6-en-2,8-diol complex: (**A**) RMSD, (**B**) RMSF, (**C**) Rg, (**D**) SASA, and (**E**) H-bond analysis.

**Figure 13 plants-12-02288-f013:**
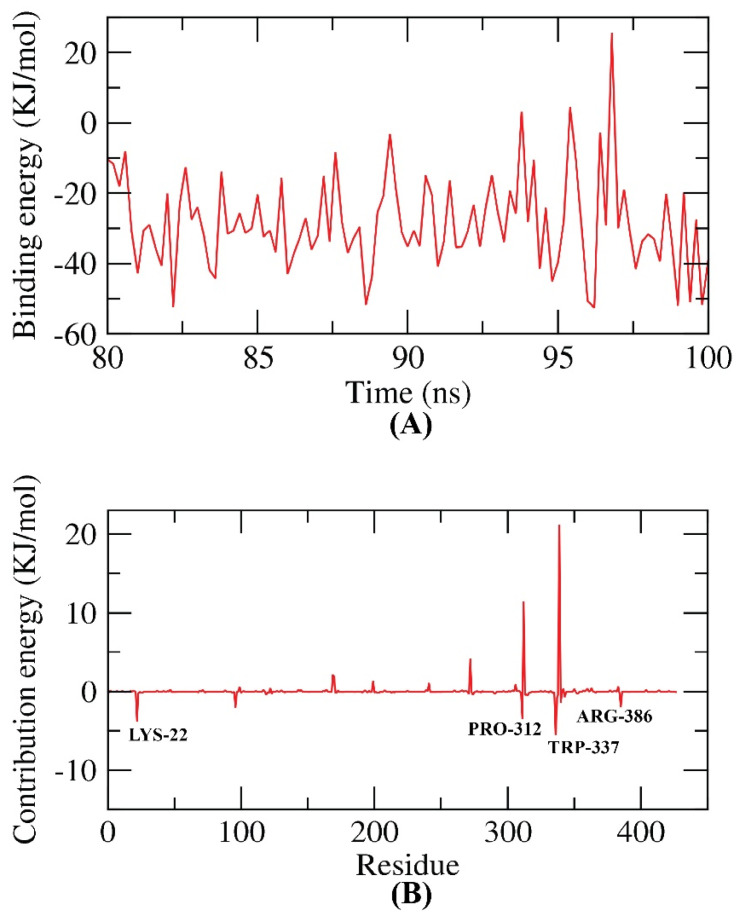
MM-PBSA study of the EPSP synthase target site–trans-p-menth-6-en-2,8-diol complex. (**A**) Total binding energy plot of trans-p-menth-6-en-2,8-diol bound with 2AAY at 100-ns time frame. (**B**) Contribution energy plot with interacting amino acids.

**Table 1 plants-12-02288-t001:** Volatile components of the essential oil of *P. crispum*.

#	Compounds	IR cal	IR ref	%	S.D.
1	α-Pinene	932	932	2.24	0.75
2	Sabinene	973	969	0.20	0.10
3	β-Pinene	979	974	1.43	0.02
4	Myrcene	991	988	5.93	0.89
5	δ-2-Carene	1016	1001	0.09	0.01
6	α-Phellandrene	1018	1002	1.15	0.02
7	ο-Cymene	1037	1022	0.27	0.22
8	Limonene	1040	1024	1.16	0.61
9	β-Phellandrene	1042	1025	15.02	2.01
10	ϒ-Terpinene	1069	1054	0.14	0.02
11	ρ-Mentha-2,4(8)-diene	1097	1085	2.73	0.91
12	ρ-Cymenene	1106	1089	3.65	0.69
13	1,3,8-ρ-Menthatriene	1119	1108	22.59	2.31
14	cis-ρ-Mentha-2,8-dien-1-ol	1138	1133	0.13	0.01
15	ρ-methyl-Acetophenone	1197	1179	0.29	0.12
16	Cryptone	1198	1183	0.21	0.09
17	Terpineol	1215	1199	0.20	0.04
18	cis-ρ-Mentha-1(7),8-dien-2-ol	1218	1227	t	0.01
19	Pulegone	1232	1233	1.50	0.12
20	Carvotanacetone	1242	1244	0.15	0.03
21	Not identified	1248	-	t	0.01
22	E-Ocimenone	1250	1235	0.19	0.06
23	Methyl octine carbonate	1300	1295	0.41	0.21
24	cis-Piperitrol acetate	1322	1332	0.39	0.11
25	trans-Carvyl acetate	1333	1339	1.32	0.98
26	Not identified	1353	-	0.10	0.03
27	trans-p-Mentha-8-thiol-3-one	1372	1371	0.12	0.09
28	trans-p-Menth-6-en-2,8-diol	1377	1371	0.10	0.01
29	Isobornyl propanoate	1399	1383	0.95	0.72
30	Not identified	1407	-	0.37	0.12
31	β-Isocomene	1410	1407	0.92	0.21
32	α-Santalene	1417	1416	0.50	0.42
33	E-Caryophyllene	1419	1417	0.62	0.31
34	γ-Muurolene	1483	1478	0.66	0.02
35	Not identified	1486	-	0.06	0.01
36	γ-Himachalene	1491	1481	0.36	0.22
37	Bicyclogermacrene	1510	1500	0.10	0.06
38	Cubebol	1524	1514	0.09	0.01
39	δ-Cadinene	1528	1522	0.38	0.21
40	Kessane	1533	1529	0.12	0.07
41	Myristicin	1535	1517	3.01	1.01
42	Elemicin	1562	1555	0.14	0.12
43	cis-Muurol-5-en-4-α-ol	1567	1559	0.44	0.12
44	6-methoxy-Elemicin	1597	1595	6.96	1.23
45	1,10-di-epi-Cubenol	1610	1618	0.12	0.06
46	γ-Eudesmol	1615	1630	t	0.01
47	Apiole	1692	1677	22.41	1.54
Monoterpene hydrocarbons (%)				56.61	
Oxygenated monoterpenes (%)				2.07	
Sesquiterpene hydrocarbons (%)				2.62	
Oxygenated sesquiterpenes (%)				0.79	
Phenylpropenes (%)				32.52	
Others (%)				4.84	
TOTAL IDENTIFIED (%)				99.45	

**Table 2 plants-12-02288-t002:** Phytotoxic parameters evaluated on *Lactuca sativa* seeds. Values are expressed as mean ± SD. * Groups are compared with the positive control, (*p* < 0.05).

Groups	Seed Germination (%)	Root Length (cm)	Hypocotyl Length (cm)	Root Length/ Stem Length
Negative control: DMSO 0.1%	96.00 ± 5.29 *	2.13 ± 0.15 *	5.50 ± 0.50 *	0.28
*P. crispum* 0.1%	27.67 ± 2.52 *	1.83 ± 0.29 *	4.17 ± 0.29 *	0.31
*P. crispum* 0.5%	15.00 ± 3.00	0.47 ± 0.06	4.67 ± 0.58 *	0.09
*P. crispum* 1%	7.67 ± 2.52	0.40 ± 0.10	3.00 ± 0.50 *	0.12
Positive control: glyphosate 1%	5.67 ± 1.15	0.09 ± 0.01	1.77 ± 0.25	0.05

**Table 3 plants-12-02288-t003:** The tested metabolites compared with a reference compound (glyphosate) and their molecular properties.

#	A-LogP	M.wt.	HBA	HBD	Rotatable Bonds	Rings	Aromatic Rings	MFPSA	Minimum Distance	Is Similar
38	3.202	222.366	1	1	1	3	0	0.08	3.11745	True
28	1.313	170.249	2	2	1	1	0	0.195	2.07929	True
27	2.491	186.314	2	1	1	1	0	0.248	2.40701	True
14	2.159	152.233	1	1	1	1	0	0.107	2.74091	True
18	2.454	152.233	1	1	1	1	0	0.11	2.78232	True
17	2.415	154.249	1	1	1	1	0	0.102	2.79147	True
25	2.779	194.27	2	0	3	1	0	0.113	3.03988	True
16	2.25	138.207	1	0	1	1	0	0.104	3.05145	True
47	2.573	222.237	4	0	4	2	1	0.154	3.0781	True
41	2.589	192.211	3	0	3	2	1	0.136	3.07947	True
24	3.043	196.286	2	0	3	1	0	0.111	3.08852	True
Glyphosate	−2.628	173.143	5	3	1	1	0	0.581	-	Reference
9	3.308	136.234	0	0	1	1	0	0	3.53118	False
10	3.448	136.234	0	0	1	1	0	0	3.55272	False
11	3.448	136.234	0	0	0	1	0	0	3.55933	False
8	3.502	136.234	0	0	1	1	0	0	3.56113	False
1	2.872	136.234	0	0	0	3	0	0	3.56803	False
3	2.926	136.234	0	0	0	3	0	0	3.57562	False
12	3.314	132.202	0	0	1	1	1	0	3.66638	False
4	3.687	136.234	0	0	4	0	0	0	3.67158	False
7	3.51	134.218	0	0	1	1	1	0	3.69333	False
31	4.131	204.351	0	0	0	3	0	0	3.75387	False
37	4.699	204.351	0	0	0	2	0	0	3.78382	False
36	4.699	204.351	0	0	0	2	0	0	3.78382	False
33	4.753	204.351	0	0	0	2	0	0	3.79333	False
34	4.798	204.351	0	0	1	2	0	0	3.7951	False
39	4.939	204.351	0	0	1	2	0	0	3.82029	False
32	4.123	204.351	0	0	3	4	0	0	3.8803	False
13	3.252	134.218	0	0	1	1	0	0	3.5249	False
6	3.254	136.234	0	0	1	1	0	0	3.52297	False
40	3.511	222.366	1	0	0	3	0	0.036	3.50425	False
2	2.926	136.234	0	0	1	2	0	0	3.59824	False
5	2.872	136.234	0	0	0	2	0	0	3.59722	False
23	3.524	168.233	2	0	7	0	0	0.122	3.59792	False
44	2.772	238.28	4	0	6	1	1	0.13	3.5717	False
22	3.054	150.218	1	0	3	0	0	0.084	3.55714	False
42	2.788	208.254	3	0	5	1	1	0.111	3.63239	False
15	2.056	134.175	1	0	1	1	1	0.107	3.5737	False
45	3.913	222.366	1	1	1	2	0	0.077	3.57082	False
46	3.86	222.366	1	1	1	2	0	0.073	3.16893	False
19	2.891	152.233	1	0	0	1	0	0.091	3.16858	False
43	3.846	222.366	1	1	1	2	0	0.077	3.15832	False
29	3.021	210.313	2	0	3	2	0	0.105	3.13708	False
20	2.697	152.233	1	0	1	1	0	0.092	3.12885	False

A-LogP: the lipid–water partition coefficient; M.wt: the molecular weight of a compound; HBA: the number of hydrogen bond acceptors in a compound, which are atoms capable of forming hydrogen bonds with other molecules; HBD: the number of hydrogen bond donors in a compound, which are atoms capable of donating hydrogen atoms for the formation of hydrogen bonds; MFPSA: the molecular fractional polar surface area, which quantifies the polar surface area of a compound relative to its total surface area; minimum distance: the shortest distance between a tested compound and a reference compound.

**Table 4 plants-12-02288-t004:** Docking analysis of the volatile components of *P. Crispum* EO against the EPSP synthase target.

Targets Screened	Tested Compounds	RMSD Value (Å)	Docking (Affinity) Score (kcal/mol)	Interactions	
				H.B	Pi -Interaction
EPSP synthase target site	*cis-ρ*-Mentha-2,8-dien-1-ol	0.80	−5.47	2	2
	Terpineol	1.23	−5.69	1	1
	*cis-ρ*-Mentha-1(7),8-dien-2-ol	1.17	−5.66	2	3
	*trans*-*p*-Mentha-8-thiol-3-one	1.69	−5.56	2	2
	*trans-p*-Menth-6-en-2,8-diol	1.26	−5.54	3	1
	Apiole	0.81	−6.85	2	3
	Glyphosate	0.19	−6.55	6	-

## Data Availability

Data will be available until request.

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
