# Peer review of "The Essential Oil of *Petroselinum crispum* (Mill) Fuss Seeds from Peru: Phytotoxic Activity and In Silico Evaluation on the Target Enzyme of the Glyphosate Herbicide"

_plants, 2023, doi:10.3390/plants12122288_

Round 1
Reviewer 1 Report
The authors studied Phytotoxic Activity and In-Silico Evaluation on the 3 Target Enzyme of the Herbicide Glyphosate of of Petroselinum crispum (Mill) Fuss Seeds 2 essential oil.
Manuscript is well written even if some corrections are necessary.
line 487: how seeds grow up in absence of light?
line 492: Please could you add formulas that you used to calculate enzymatic activity? is not clear.
Author Response
The authors studied Phytotoxic Activity and In-Silico Evaluation on the 3 Target Enzyme of the Herbicide Glyphosate of Petroselinum crispum (Mill) Fuss Seeds 2 essential oil.
Manuscript is well written even if some corrections are necessary.
- line 487: how seeds grow up in absence of light?
R1: Thank you for your comments, the seeds grow up in absence of light, because this stage needs these conditions to germinate and simulate the seeds under the ground. According to the protocol by Viera et al. we could germinate our seeds under a photoperiod of 12 h light and 12 h of darkness. However, we rewrite this sentence to avoid a mistake in interpretation.
and we added in the methodology.
- line 492: Please could you add formulas that you used to calculate enzymatic activity? is not clear.
R2: Thank for your comments, the formulas were added in method section.
Reviewer 2 Report
In the present study, Oscar et. al. have used experimental and computational techniques to investigate the Phytotoxic Activity of the Essential Oil of Petroselinum crispum (Mill) Fuss Seeds 2 from Peru. My comments for authors are below
- In the figure write separately like A), B), and C) and mention what they show.
- Authors should mention EPSP synthase in the introduction section, focusing on structure details and also active site residues and their roles.
- 3.6 and 3.7 section, the authors have not cited any reference here all the software and PDB references should be cited
- for how long NVT and NPT equilibration was performed mention it.
- mention in detail what parameters were used for a production run for instance when the integration algorithm was used, integration time, etc.
Minor English can be improved.
Author Response
In the present study, Oscar et. al. has used experimental and computational techniques to investigate the Phytotoxic Activity of the Essential Oil of Petroselinum crispum (Mill) Fuss Seeds 2 from Peru. My comments for authors are below
- In the figure write separately like A), B), and C) and mention what they show.
R1: We have improved the description of each figure.
- Authors should mention EPSP synthase in the introduction section, focusing on structure details and also active site residues and their roles.
R2: Thank you for your comments, we added the following information in the introduction:
EPSP synthase, or 5-enolpyruvylshikimate-3-phosphate synthase, is a crucial enzyme involved in the shikimate pathway, which is responsible for the biosynthesis of aromatic amino acids in plants, bacteria, and some parasites. Within the active site of EPSP synthase, several key residues play essential roles in catalysis and substrate binding. One of the critical residues is a conserved Aspartate residue (Asp) and lysine are present, which aids in substrate binding and stabilizes the transition state during the reaction. These active site residues work cooperatively to ensure efficient catalysis and precise substrate specificity, making EPSP synthase an attractive target for the development of herbicides and antimicrobial agents.
- 6 and 3.7 section, the authors have not cited any reference here all the software and PDB references should be cited
R3: thank you for your comments, it was cited.
- for how long NVT and NPT equilibration was performed mention it.
R4: The systems were then equilibrated using NVT ensembles (constant number of particles, volume, and temperature) and NPT ensembles (constant number of particles, pressure, and temperature), both of which were carried out at 300 K and 1 atm for 100 picoseconds (ps). The thermostat and barostat for the simulation that was run, respectively, were chosen to be the V-rescale and Parrinello-Rahman. Finally, the production run was carried out for 100 nanoseconds (ns) at 300 K with a 2 fs time step.
- mention in detail what parameters were used for a production run for instance when the integration algorithm was used, integration time, etc.
R5: The thermostat and barostat for the simulation that was run, respectively, were chosen to be the V-rescale and Parrinello-Rahman. Finally, the production run was carried out for 100 nanoseconds (ns) at 300 K with a 2 fs time step and it was included in the methodology.
Reviewer 3 Report
The manuscript, entitled " The Essential Oil of Petroselinum crispum (Mill) Fuss Seeds from Peru: Phytotoxic Activity and In-Silico Evaluation on the Target Enzyme of the Herbicide Glyphosate " This work is merited for publication in Plants after some major modification. So, I have some points that may help to improve the work as follows:
1-Abstract is good but need more explain about the main aim of work
2- The introduction should be extended to discuss the hypothesis and research questions in details. Additionally, the introduction should cover the recent literature related to this subject.
3- Material and methods
The methodologies should be explained in details so that the results are reproducible.
4-Results
The results are clear and important.
5-Discussion
The discussion section still needs improvement, and should be linked to the findings of the previous reports on this topic.
6- The conclusion
A section for conclusions need more explain and should include the most significant findings and future works only.
7- English writing should be checked by a native English-speaking expert.
English writing should be checked by a native English-speaking expert.
Author Response
REVIEWER 3
The manuscript, entitled " The Essential Oil of Petroselinum crispum (Mill) Fuss Seeds from Peru: Phytotoxic Activity and In-Silico Evaluation on the Target Enzyme of the Herbicide Glyphosate " This work is merited for publication in Plants after some major modification. So, I have some points that may help to improve the work as follows:
1-Abstract is good but need more explain about the main aim of work
R1: The aim was corrected according to your suggestion.
2- The introduction should be extended to discuss the hypothesis and research questions in details. Additionally, the introduction should cover the recent literature related to this subject.
R2: The references were updated and we added a paragraph about the EPSP synthase in the introduction.
3- Material and methods
The methodologies should be explained in details so that the results are reproducible.
R3: Thank you for your comment, we explained the formulas used in the antioxidant markers.
4-Results
The results are clear and important.
R4: Thank you for your comments.
5-Discussion
The discussion section still needs improvement, and should be linked to the findings of the previous reports on this topic.
R5: According to your suggestions, we improved our discussion.
6- The conclusion
A section for conclusions needs more explain and should include the most significant findings and future works only.
R6: Thank you for your suggestions, we improved the redaction our article.
7- English writing should be checked by a native English-speaking expert.
R7. English was improved within the manuscript.
Thank you for your comments, some changes were highlighted with pink and yellow colors.
Round 2
Reviewer 2 Report
The authors have made changes suggested by me, therefore, I recommend the publication of this manuscript in its present form.
Reviewer 3 Report
Authors have suitably revised the manuscript by addressing the reviewer comments and suggestions. This can be accepted for publication.
Minor editing of English language required